# Fast Path Planning for Polar Surface Unmanned Vessels Based on GI-ACO-A* Algorithm

1st Zilong Qu
*Merchant Maritime College*
*Shanghai Maritime University*
Shanghai, China
202330110093@stu.shmtu.edu.cn

2nd Xiaojun Mei
*Merchant Marine College*
*Shanghai Maritime University*
Shanghai, China
xjmei@shmtu.edu.cn

3rd Huafeng Wu*
*Merchant Marine College*
*Shanghai Maritime University*
Shanghai, China
hfwu@shmtu.edu.cn

4th Hongde Qin
*Qingdao Innovation and Development Base*
*Harbin Engineering University*
Qingdao, China
qinhongde@hrbeu.edu.cn

5th Kun Zhang
*Merchant Marine College*
*Shanghai Maritime University*
Shanghai, China
zhangkunk1@163.com

*Abstract*—Traditional A* algorithms exhibit two deficiencies in the realm of surface unmanned vessel path search:The first deficiency in path search is the excessive redundancy of visited nodes, leading to a high number of unnecessary node accesses that markedly degrades the efficiency of the path-finding process;The second deficiency is the neglect of the unmanned surface vessel's profile in the path planning.To enhance path search efficiency, the GI-ACO-A* algorithm (Goal-Induced A* under Ant Colony Optimization influence) is proposed, along with a navigation strategy integrating global path planning with local obstacle avoidance for unstructured environments. This approach features a global layer for overview path planning and a local layer for real-time obstacle evasion, adapting to the complex and dynamic surface conditions.Furthermore, this study processes radar imagery to obtain a sea ice distribution map and subsequently filters the radar image data to facilitate the modeling of a gridded environment. Simulation tests have validated its navigation safety and efficiency under complex surface conditions.

*Index Terms*—path planning, GI-ACO-A* algorithm, goal-induced, obstacle avoidance, usv

## I. Introduction

Unmanned surface vessels (USVs) have consistently been a research focus in recent years, particularly in the areas of route planning, environmental perception, and distributed formation control. Their autonomous navigation technology enables operation in harsh environments and dangerous zones, maximizing the reduction of personnel casualties and ensuring the safety and efficient operation of USVs.

In recent years, in the realm of USV path planning and obstacle avoidance, Bai [1] have proposed a Phototropism-inspired Gradient Routing (PGR) algorithm. This method guides USVs to navigate around obstacles and reach the target point by utilizing the intensity of light. Liu X [2] have introduced a methodology for Unmanned Surface Vehicle (USV) dynamic obstacle avoidance and path planning, which is grounded in the Ant Colony Algorithm (ACA) and Clustering Algorithm (CA). This approach is tailored for intricate maritime environments, capable of adaptively identifying the

complexity of local static and dynamic obstacles at sea. Furthermore, it accomplishes large-scale dynamic path planning and obstacle evasion in maritime contexts through the integration of multi-source information. Yao P [3] utilized the Biased Min-Consensus (BMC) protocol for optimal path planning, factoring in travel time and risk assessment to ensure USV obstacle avoidance and swift target arrival. MahmoudZadeh [4] developed a non-interfering continuous path planning system for multiple USVs, designed to facilitate efficient collaborative operations in complex marine sampling missions. The system, based on an innovative B-Spline data framework, integrates Particle Swarm Optimization (PSO) as its solving engine to ensure optimality, smoothness, and task-compliance of the generated path curves among multiple USVs. Yan-Li Chen [5] proposed a hybrid path planning method that integrates an enhanced Ant Colony Optimization (ACO) with the Artificial Potential Field (APF) algorithm, aiming to determine the globally optimal path from start to goal for Unmanned Surface Vehicles (USVs) in a gridded environment while effectively avoiding unknown obstacles during navigation. By refining the ACO algorithm to enhance its path search capabilities in complex environments and improving the APF algorithm to boost the USV's real-time obstacle avoidance in dynamic navigation, the hybrid approach optimizes comprehensive path planning performance while ensuring operational safety and efficiency. This research significantly contributes to efficient pathfinding and dynamic obstacle avoidance for USVs; however, it has not fully addressed the impact of the USV's hull profile and kinematic constraints on navigation performance. In the field of USV navigation, conventional path planning and obstacle avoidance strategies, often based on global or local navigation maps, overlook the practical limitations imposed by the hull's shape and size. For instance, factors such as the USV's beam, length, and draft critically affect maneuverability and passage in complex hydrological conditions.

In the realm of path planning algorithm research, grid-based

search methods have been widely adopted in autonomous navigation and robotics due to their practicality, simplicity, and capability for analytical optimality. These algorithms are primarily categorized into two types: deterministic and non-deterministic.

Deterministic algorithms, such as A* [6], Dijkstra [7], D* [8], and RRT* [9], primarily involve searching for valid paths from a start to a goal in a given environment, yielding consistent output results. In contrast, non-deterministic algorithms like GA [10], AC [11], SA [12], and PSO [13]employ probabilistic and heuristic strategies to explore the search space, aiming to achieve near-optimal solutions within acceptable time and resource constraints.

The A* algorithm, introduced by Hart in 1968, combines the core features of Dijkstra's algorithm with heuristic search strategies, significantly enhancing search efficiency in path finding and graph traversal problems. It has since become a widely used method for addressing issues in artificial intelligence, robotic path planning, network routing, and navigation. Its versatility lies in the customizable design of the heuristic function, allowing A* to be adapted for various search scenarios in recent research. Tang G [14] proposed a geometric A* algorithm-based path planning method that employs a filtering function to refine the paths generated by A* algorithm, thereby preventing excessive turning angles and irregular paths, ensuring the continuity of speed and acceleration during AGV movement. Song R [15] introduced a smoothed A* algorithm that enhances path performance using three path smoothers to reduce unnecessary reversals, eliminate redundant waypoints, and provide a more continuous trajectory. Sang H [16] proposed the MTAPF local path planning algorithm, which references an improved heuristic A* algorithm to generate a globally optimal path and subdivides it into multiple sub-goals, forming a sequence of sub-target points. Dengxing Zhang [17] proposed a combination of APF and A* algorithm, incorporating the repulsive force of obstacles at turning points as a penalty term in the evaluation function, thereby moving turning points away from obstacles to enhance AGV control safety. Studies [18]- [21] and those mentioned have extensively researched and innovated in reducing turning angles and obstacle avoidance. However, as the problem scale increases, there has been insufficient research on the decline in search efficiency due to excessive redundancy in A* algorithm search nodes.

As problem scales expand, non-deterministic algorithms demonstrate significant advantages in rapidly converging on a relatively good path. For instance, Yanli Chen [22] proposed an enhanced Ant Colony Optimization-Artificial Potential Field (ACO-APF) hybrid algorithm, incorporating an adaptive early warning mechanism to enhance the efficiency of local and global path planning and navigation safety for USVs in dynamic environments. Literature [23] presents an improved Ant Colony Optimization algorithm integrated with Fuzzy Logic (ACO-FL) for local path planning considering wind, currents, waves, and dynamic obstacles. Literature [24] introduces a novel pheromone updating method based on obstacle distance

to optimize path planning. Literature [25] extends the ACO algorithm by integrating Mamdani and Takagi-Sugeno-Kang (TSK) fuzzy inference systems to address multi-objective optimization problems. Literatures [22]- [25] all refine the Ant Colony Intelligence optimization algorithm, ensuring USV safety by avoiding obstacles under various conditions. Literatures [26]- [27] improve the Particle Swarm Optimization algorithm, enhancing the efficiency of finding the optimal path, and respectively employ the Velocity Obstacle (VO) method, considering USV dynamics, environmental constraints, and generating real-time obstacle avoidance paths. Literatures [28]-[30] utilize the Ant Colony Optimization algorithm to produce suboptimal paths, incorporating them as part of the initial population in Genetic Algorithms to accelerate convergence. Although non-deterministic heuristic algorithms can achieve globally optimal solutions in small-scale scenarios, their slower convergence in large-scale scenarios can lead to premature convergence and the generation of suboptimal paths.

In summary, both deterministic and non-deterministic algorithms have their strengths but face challenges as problem scales increase. Researchers have made notable innovations and achievements in optimal path generation and obstacle avoidance, yet there is a research gap in improving algorithmic efficiency with scale. Therefore, to enhance search efficiency and ensure the operational safety of USVs in polar environments, with sufficient buffer zones between the USV and ice obstacles for smoother paths, we propose a path planning method based on GI-ACO-A* algorithm and radar imagery. Overall, the main contributions of this paper are as follows:

- By refining the Emperor Penguin Algorithm with K-Means to process radar imagery and obtain sea ice distribution maps, followed by information filtering of the radar images to facilitate grid-based environmental modeling.
- The pheromone update mechanism of the Ant Colony Optimization (ACO) algorithm is enhanced for preprocessing the modeled graphics, and the weight distribution during the next node selection in the A* algorithm is adjusted. Moreover, guidance is applied to the node selection process based on the target direction for the USV's destination, optimizing path planning performance.
- A navigation strategy that integrates global path planning with local obstacle avoidance is proposed to adapt to the complex and variable polar water conditions, where the global layer handles overarching path planning, and the local layer focuses on real-time obstacle evasion.

## II. ENVIRONMENTAL MODELING

After acquiring radar imagery ("Fig. 1"), a crucial step is to analyze the image to identify and extract regions of interest (ROI). In radar image analysis, circular areas are typically designated as ROIs, particularly in the monitoring and interpretation of ice navigation scenarios. These circular regions may contain vital information such as sea ice distribution, ice thickness, and ice ridge characteristics, which are essential for understanding polar environments, ensuring maritime safety,

and supporting scientific research. Consequently, the precise identification and extraction of these ROIs are central to radar image processing, necessitating the use of efficient and robust image segmentation and target detection algorithms to ensure the extracted areas provide an accurate foundation for subsequent image analysis and data interpretation [31].

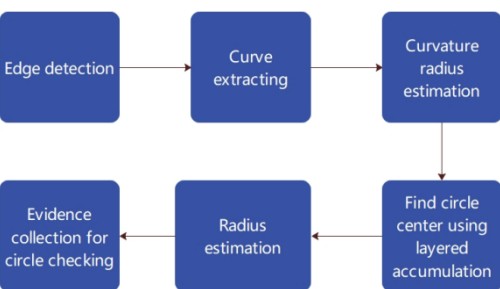

Fig. 2. Flow diagram of the CACD algorithm.

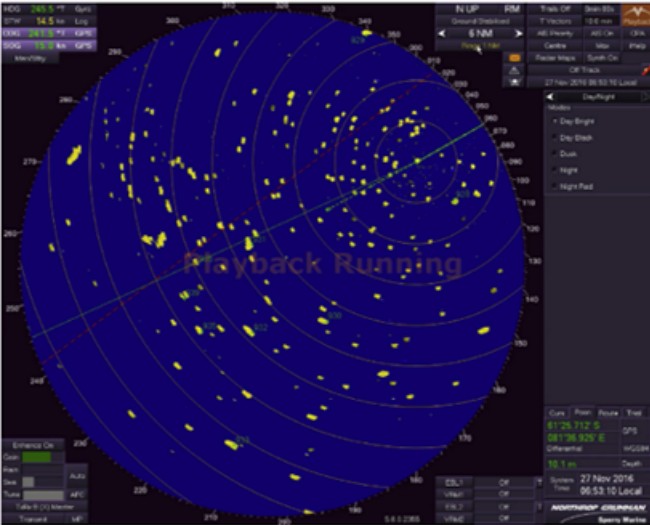

Fig. 1. Radar imagery of sea ice distribution.

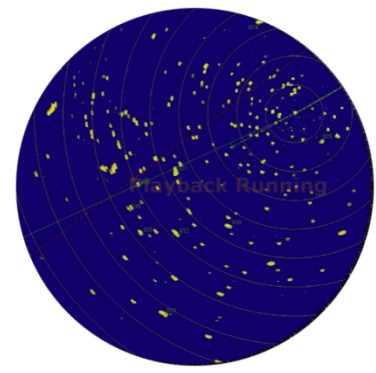

Fig. 3. Flow diagram of the CACD algorithm.

### A. Central Circular ROI Extraction and Unknown Obstacle Detection in Radar Imagery

Utilizing the Curvature-Assisted Circle Detection (CACD) algorithm [32] for identifying circular regions in radar imagery, which pre-estimates curvature to reduce global point accumulation and minimize multi-scale interference, significantly enhancing circle detection efficiency and accuracy."Fig. 2" illustrates the flowchart of the CACD algorithm, while "Fig. 3" shows the central circular ROI obtained using the CACD algorithm. The following sections outline the key steps of the CACD algorithm.

The extracted curve formula, as expressed in "equation (1)",represents a curve composed of n points with coordinates $(x_0, y_0), (x_1, y_1), ..., (x_n, y_n)$.

$$L_i = \left[ \begin{array}{cccc} x_0 & x_1 & \dots & x_n \\ y_0 & y_1 & \dots & y_n \end{array} \right] \tag{1}$$

Curvature is calculated using "equation (2)". $(u, \sigma)$ is a 2D Gaussian window centered at $u$ with deviation $\sigma$.Where $\dot{X}(u, \sigma) = x(u) * \dot{g}(u, \sigma)$, $\ddot{X}(u, \sigma) = x(u) * \ddot{g}(u, \sigma)$, $\dot{Y}(u, \sigma) = y(u) * \dot{g}(u, \sigma)$, $\ddot{Y}(u, \sigma) = y(u) * \ddot{g}(u, \sigma)$, $g(u, \sigma)$, $f * g$ represent the first and second derivatives of $g(u, \sigma)$, respectively.

$$c(u, \sigma) = \frac{\dot{X}(u, \sigma)\ddot{Y}(u, \sigma) - \ddot{X}(u, \sigma)\dot{Y}(u, \sigma)}{\left(\dot{X}(u, \sigma)^2 + \dot{Y}(u, \sigma)^2\right)^{1.5}} \tag{2}$$

The final radius estimate is given by "equation (3)":

$$r(u, \sigma) = \begin{cases} \frac{1}{c(u,\sigma)} & \frac{2}{3m} < c(u, \sigma) < 1 \\ 0 & otherwise \end{cases} \tag{3}$$

To accurately extract sea ice information and eliminate interference from non-sea-ice image colors, an improved penguin optimization algorithm-based multi-threshold image segmentation method [33] is employed to identify and retain regions matching sea ice color characteristics while filtering out other color information. The filtered image is shown in "Fig. 4".

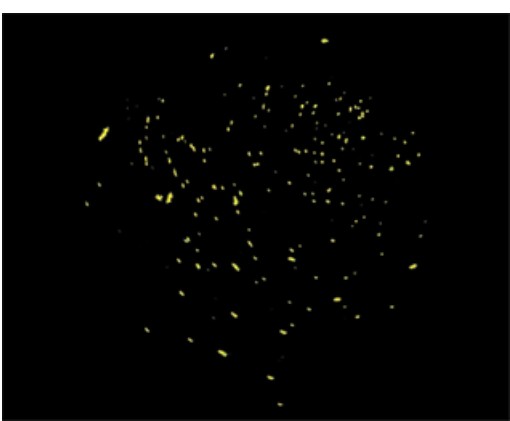

Fig. 4. Sea ice distribution filtering map in radar imagery.

### B. Rasterization Processing of Sea Ice Distribution Filter Map

As shown in "Fig. 4", the filtered image exhibits more distinct color regions compared to "Fig. 3". However, the application of the improved Penguin Optimization Algorithm (refer to [33]) in this study's imagery led to premature

convergence, leaving some pixels (including other interfering points) unfiltered. To address this, a K-means-based RGB clustering classification algorithm is proposed on the basis of the aforementioned experimental results to further enhance the filtering of non-sea-ice distribution pixels.

The RGB color model is an additive color model where R stands for red, G for green, and B for blue. In digital devices, RGB color is commonly represented by three 8-bit channels, each ranging from 0 to 255, thus the RGB color formula can be expressed as:$\mathrm{RGB}(\overline{R}, \overline{G}, \overline{B})$

- 1)Traverse the pixels of the preliminarily filtered image to obtain the top 4 clusters with the most identical RGB classifications (see Table 1 for the dataset in this study), labeled as $a = \{a_1, a_2, ..., a_{m-1}, a_m\}$, $b = \{b_1, b_2, ..., b_{n-1}, b_n\}$, $c = \{c_1, c_2, ..., c_{x-1}, c_x\}$, where each set has equal RGB values. Remaining pixels with distinct RGB values are grouped into set $t = \{t_1, t_2, ..., t_{l-1}, t_l\}$ for subsequent classification.
- 2)Calculate the distance between data points (RGB values of each element in set t) and the centroid points (RGB cluster set values from Table 1) using "equation (4)" for all elements in set t.Distances to $d(a, t_l)$, $d(b, t_l)$, $d(c, t_l)$;Set a threshold $\theta$ (a constant, adjustable based on the context); if $u = min(d(a, t_l), d(b, t_l), d(c, t_l)) < \theta$, then select $t_l$ from $u$ and add to the respective set,otherwise continue the traversal.
- 3)Update set $a, b, c$ with the mean $\mathrm{RGB}(\overline{R}, \overline{G}, \overline{B})$ values calculated by "equation (5)" and refresh set (as shown in Table 1).Further update o using "equation (6)", where $count(x)$ denotes the count of elements in set $x$.Repeat steps 2) and 3) until set $t = \emptyset$ or the maximum number of iterations is reached.

TABLE I
RGB CLUSTER SETS

| Set | a | b | c | t |
|---|---|---|---|---|
| RGB | (215,210,39) | (0,0,0) | (51,27,82) | other |
| Count | 1836 | 196940 | 257 | 2568 |

$$d(\mathbf{x}, \mathbf{c}) = \sqrt{\sum_{i=1}^{n} (x_i - c_i)^2} \tag{4}$$

$$\overline{x} = \frac{\sum_{i=1}^{n} (x_i)}{n} \tag{5}$$

$$\theta = (1 + \frac{count(t)}{count(a) + count(b) + count(c)}) \bullet \theta \tag{6}$$

The data variation process in this study (shown in "Fig. 5") clearly illustrates the progressive purification effect of the algorithm on the dataset. With each iteration, the dataset's purity increases, eventually flattening out after a certain number of iterations, indicating the desired level of purification has been achieved. The top two sets with the most elements are retained

and color-transformed, as shown in "Fig. 6", which presents the final filtered image results.

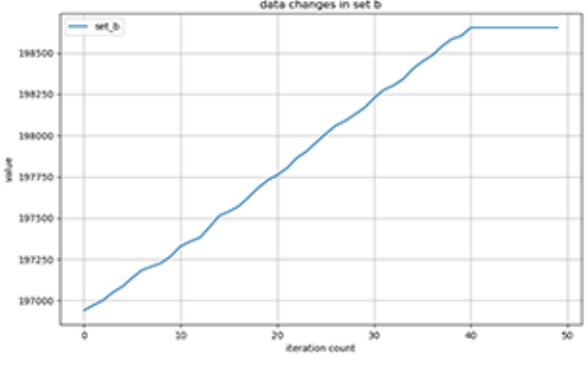

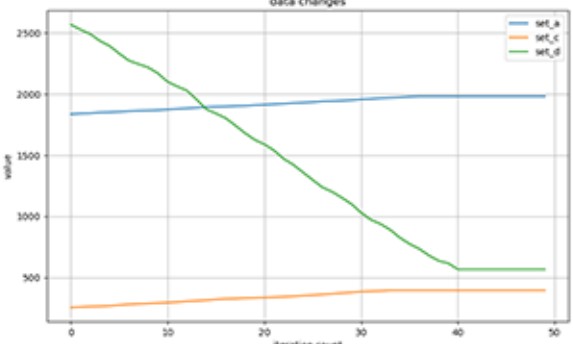

Fig. 5. Data variation graph of sets $a, b, c, d$.

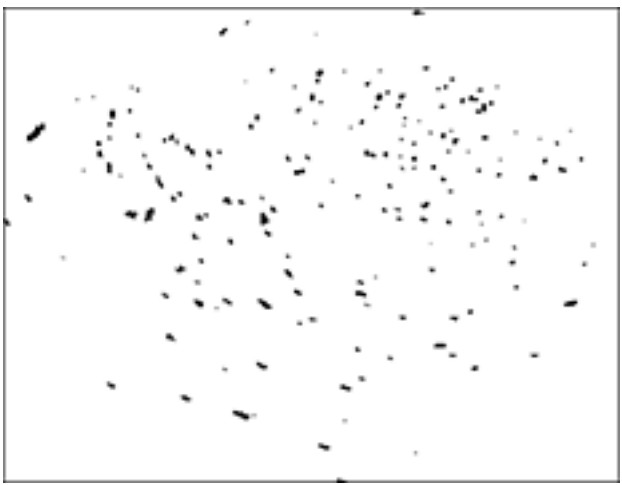

Fig. 6. Final result image of image filtering.

In the field of path planning, rasterization is widely used to abstract continuous space into a discrete set of units, reducing problem complexity and simplifying algorithm implementation. This approach, by dividing space into a series of regular cells, provides a limited and structured search space for pathfinding algorithms [34]. Rasterization facilitates direct integration of environmental information, and in this paper,

by mapping filtered radar image pixels to elements in a 2D matrix (as shown in "Fig. 7"), pixels are classified as safe or obstacle areas based on RGB values, constructing a map with environmental obstacle information. This method not only enhances path planning efficiency but also ensures algorithm accuracy and reliability.

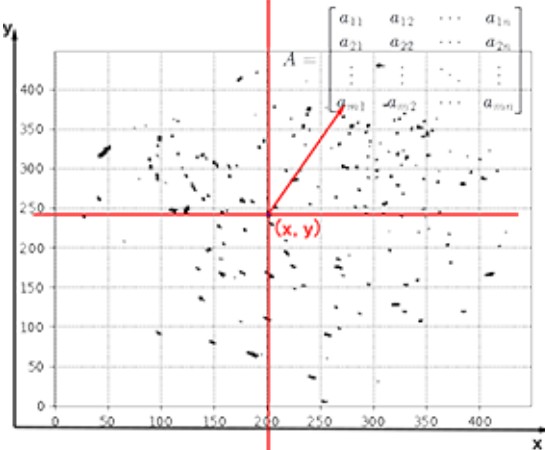

Fig. 7. Sea ice distribution grid map.

## III. GI-ACO-A* ALGORITHM

In the field of path planning, the ACO [11] algorithm seeks the optimal path from start to end by simulating ants depositing pheromones along routes. However, as the problem scale increases, the convergence rate of ACO may degrade, particularly in high-dimensional search spaces [35].

The A* algorithm is a prevalent heuristic search technique for path planning and graph traversal [6], with the challenge of selecting an appropriate heuristic function in complex scenarios critically affecting its efficacy.

This chapter first validates the pivotal role of the positive feedback mechanism of pheromone concentration in the Ant Colony Optimization (ACO) algorithm for path convergence. Subsequently, the pheromone update formula in the ACO algorithm is optimized, and the refined strategy is integrated into the A* algorithm to enhance overall efficiency. Finally, by guiding the search path of the A* algorithm with the target heading of Unmanned Surface Vehicles (USVs) and pruning non-essential visit nodes, the search efficiency of the algorithm is further improved.

### A. ACO Algorithm

In path planning for complex graphs, the ACO algorithm often exhibits slow convergence rates and high computational resource consumption during early stages due to insufficient initial pheromone accumulation. In contrast, GAs may converge more rapidly early on, potentially harboring superior genetic information in the initial population. To assess the impact of pheromone concentration in ACO on node selection in path planning within GAs, this study constructed a simplified simulation diagram ("Fig. 8") to reduce complexity and conducted

experimental analysis. In "Fig. 8", black denotes obstacles, white indicates navigable areas, light blue squares represent the starting position, and dark blue indicates the destination. A single gene in the Genetic Algorithm is composed of a list $list = [(x_0, y_0), (x_1, y_1), ....., (x_n, y_n)]$, where each tuple represents the coordinate position of a node on a 2D map, with n denoting the number of nodes in the list. $Fitness$ is calculated using "equation (7)", where $N$ represents the number of tuples in the list, $x_i, y_i$ denote the 2D coordinate positions of the path nodes within the gene sequence, and $listArray = []$ stores $M$ gene sequences.

$$fitness = \frac{1}{\sum\limits_{i=1}^{N-1} \sqrt{(x_{i+1} - x_i)^2 + (y_{i+1} - y_i)^2}} \qquad (7)$$

- 1)Initially, genes are initialized, and during pheromone initialization, paths corresponding to the start, destination, and obstacle nodes are not selected, setting their pheromone levels to 0. For other paths, assuming equal selection probabilities at the algorithm's onset, their pheromone levels are initialized to 1. This setup ensures equal exploration opportunities for all paths at the beginning, aligning with the balance between exploration and exploitation in genetic algorithms.

- 2)During gene mutation, a random parameter $x \in (0, 1)$ is set, and "equation (8)" is used to determine if the current gene sequence $listArray[i]$ is selected. In this formula, parameter $a$ is adjusted according to the problem size, typically selecting the range of $a \in (0, 0.5)$ to ensure the randomness of gene mutation. For selected gene sequences, there is a tendency to choose units with higher pheromone concentrations as the next node based on the pheromone concentration map, due to the positive feedback mechanism it provides. This mechanism not only enhances the directionality of the search but also accelerates the convergence of the algorithm, with the newly mutated genes stored in $newListArray = []$ for algorithmic iteration.

$$\text{random}(x) = \begin{cases} Check, & x \leq a \\ \text{Uncheck}, & x > a \end{cases} \qquad (8)$$

- 3)To hasten convergence, this study employs the Roulette Wheel Selection Algorithm [36] to preferentially select gene sequences with shorter path lengths, or higher fitness, for crossover and inversion operations.The new sequences generated from crossover or inversion are incorporated into $newListArray = []$. This strategy not only enhances genetic diversity within the population but also promotes convergence to the optimal solution by reinforcing the inheritance of favorable genes.

- 4)Gene sequences are randomly culled from list $listArray[]$ using the Tournament Selection algorithm [37], with $N = len(newListArray)$ sequences removed and the remainder $newListArray[]$ merged into $listArray[]$. The pheromone concentration table is updated based on the gene sequences in the new generation

$listArray[]$. The pheromone concentration update is performed using "equation (9)".

$$\tau(N_{G+1}) = (1 - \rho) \times \tau(N_G) + \sum_{i=1}^{K} \Delta\tau_i(N) \quad (9)$$

$\tau(N_{G+1})$ represents the pheromone concentration in cell $N$ of generation $G$, $\rho$ is the evaporation coefficient, $K$ is the number of genes, and $\tau_i(N)$ is the pheromone increment left by the $i-th$ generation gene individual in cell $N$ of generation $G$. The pheromone increment is as shown in "equation (10)".

$$\Delta\tau(N) = \frac{Q}{1/F} \quad (10)$$

$Q$ represents the total pheromone left by a gene individual upon leaving a cell, and $F$ denotes the fitness of the gene individual. From Formulas (7) and (9), it can be inferred that the pheromone increment is directly proportional to the inverse of the path length. That is, the shorter the path length, the greater the pheromone increment, leading to more pheromone accumulation on shorter paths. This relationship indicates that shorter paths are likely to receive more pheromone during updates, making them more probable to be selected during genetic algorithm mutations. Node selection probability during mutation operations is determined according to "equation (11)".

$$P(N_i) = \frac{(\tau(N_i))^\alpha \times (\eta(N_i))^\beta}{\sum_{N_j \in accessible cells} (\tau(N_j))^\alpha \times (\eta(N_j))^\beta} \quad (11)$$

$N_i$ denotes the set of selectable next cells around the current node, $\tau(N_i)$ represents the pheromone concentration of cell $N_i$, and $\eta(N_i)$ is the reciprocal of the distance from the current cell $N_i$ to the target. Parameters $\alpha$ and $\beta$ are used to quantitatively adjust the relationship between pheromone concentration (a signal left on paths to guide subsequent searches) and heuristic evaluation (an immediate assessment of solutions based on problem-specific knowledge), optimizing the algorithm's balance between exploration of unknown areas and exploitation of known information. This balance is crucial for the algorithm's effectiveness and convergence.

- 5)The procedure is repeated until the maximum iteration count is reached, yielding the $fitness$ genetic sequence. This study employs a maximum of 120 iterations, with the information concentration map reviewed every 40 generations to visualize the convergence trajectory "Fig. 9". The final optimal path aligns with the ultimate unit concentration update of the information concentration.

### B. ACO-A* Algorithm

The findings indicate that in ant colony intelligence optimization algorithms, the distribution of pheromone concentration plays a positive feedback role in guiding the ants'

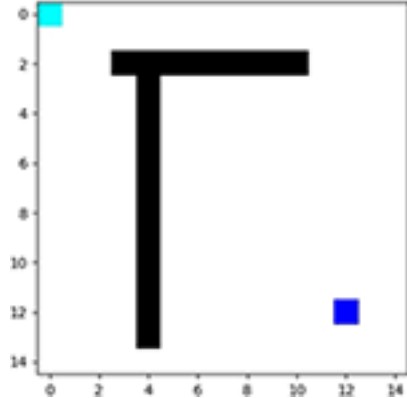

Fig. 8.  Simplified barrier simulation diagram.

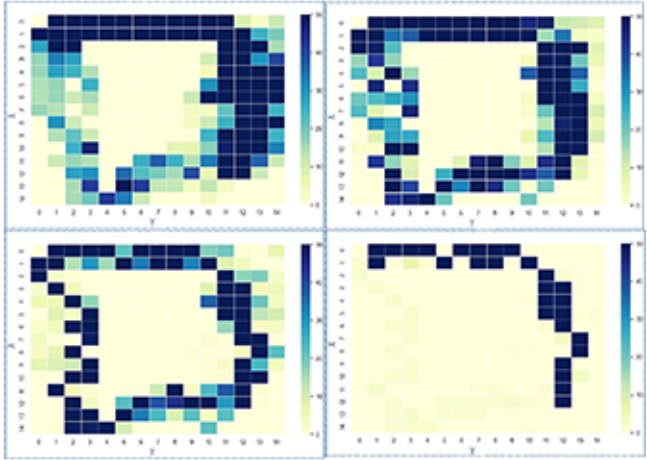

Fig. 9.  Pheromone concentration change chart

decision-making process for selecting the next node. By incorporating the initial pheromone concentration from the ant colony intelligence algorithm as a guiding factor into the cost function of the A* algorithm, its performance can be enhanced, thereby improving its efficiency in path search tasks. The iteration of initial pheromone concentration did not significantly affect the search efficiency of the A* algorithm. Consequently, this paper adjusts the pheromone update mechanism based on the characteristics of the A* algorithm (as outlined below) and constructs a model map to rigorously analyze the efficacy of the updated pheromone concentration formula in improving the efficiency of the A* search algorithm.

- In-depth analysis reveals that the A* search algorithm stores fewer nodes in environments with uniformly distributed obstacles compared to those with unevenly distributed obstacles, indicating enhanced efficiency in reaching the target point.
- In graph theory and path planning, the shortest path between two points, based on Euclidean geometry and ignoring obstacles, typically follows a straight line $l$. When obstacles are present, although the direct line may be obstructed, the principle of the shortest path still

applies in the obstacle-free regions near the line. In such cases, even when the path must detour around obstacles, the efficiency of the indirect route is usually highest near the direct line. The adjustments to "equation (10)" based on the aforementioned research are as follows:

$$\Delta \tau(N) = \frac{Q}{1/F} + C(N)^{\alpha} + D(N)^{\beta} \qquad (12)$$

Where $C(N)^a$ represents the number of obstacles surrounding node $N$, and $\alpha$ is a constant. Adjusting the weight of $\alpha$ can effectively alter the node's relative importance within the overall environment. $D(N)$ denotes the distance from the current node's coordinates $(x_0, y_0)$ to the line $l$ (the straight line connecting the start and end points: $Ax + By + C = 0$), as shown in "equation(13)".After the dynamic pheromone concentration update process, the network structure was preprocessed to optimize the distribution of pheromones. The post-processed pheromone concentration distribution, as depicted in "Fig. 10(a)", reveals the spatial variation characteristics of pheromone concentration within the network. During path search, the A* algorithm evaluates node priority based on heuristic and actual distance information, with the influence of pheromone concentration from ant colony optimization affecting priority judgments. These nodes are closer to the target or possess superior path quality. In Figure 10, "Fig. 10(b)" illustrates the path output of the traditional A* algorithm, while Figure "Fig. 10(c)" displays the ACO-A* algorithm's results. The yellow areas in "Fig. 10(b)" and "Fig. 10(c)" indicate the nodes traversed by each algorithm. Comparison reveals that the ACO-A* algorithm not only visits fewer nodes but also generates paths with superior smoothness and efficiency compared to the conventional A* algorithm.

$$D(N) = \frac{|Ax_0 + By_0 + C|}{\sqrt{A^2 + B^2}} \qquad (13)$$

### C. GI-ACO-A* Algorithm

In the traditional A* algorithm, when searching for the next node, the cost function must be calculated for all adjacent nodes. However, based on the direction and position of the target, certain nodes' cost functions do not require evaluation. As depicted in Figure "Fig. 11", with the unmanned surface vessel at position $(x_{start}, y_{start})$, accessible nodes include $F = \{f_1, f_2, f_3, f_4, f_5, f_6, f_7\}$, and the target node is at position $(x_{end}, y_{end})$. Following the specified steps, there are four resultant scenarios, denoted as $A \cup C, A \cup D, B \cup C, B \cup D$.

- If $x_{end} > x_{start}$, nodes in set $F$ that meet the criteria are $A = \{f_1, f_2, f_3\}$.
- If $x_{end} < x_{start}$, nodes in set $F$ that meet the criteria are $B = \{f_5, f_6, f_7\}$.
- If $y_{end} > y_{start}$, nodes in set $F$ that meet the criteria are $C = \{f_1, f_7\}$.
- If $y_{end} < y_{start}$, nodes in set $F$ that meet the criteria are $D = \{f_3, f_4, f_5\}$.

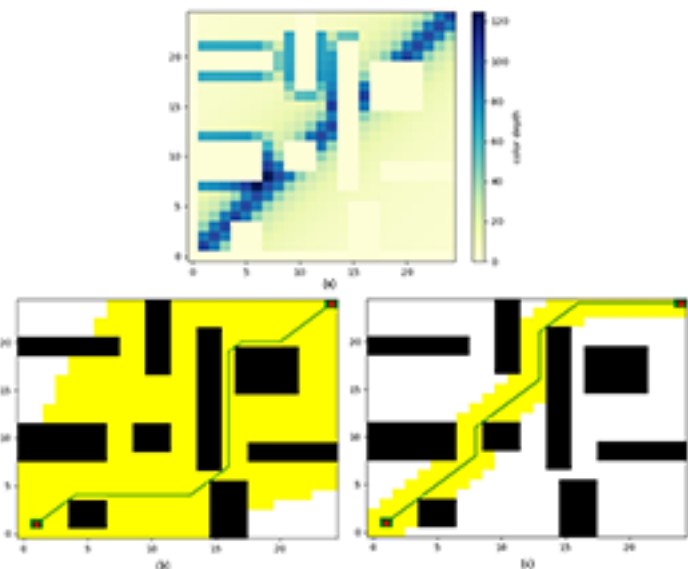

Fig. 10. (a):Pheromone concentration map,(b):A* algorithm results graph,(c):ACO-A* algorithm results graph

Comparing two improved A* algorithms: ACO-A* and GI-ACO-A*, experimental analysis (as shown in "Fig. 12") reveals that the GI-ACO-A* algorithm exhibits higher efficiency in searching for optimal solution paths. Specifically, the GI-ACO-A* algorithm significantly reduces the number of nodes visited during optimal path search (yellow areas denote visited nodes), indicating that the introduced enhancements effectively decrease the exploration of search space, thereby accelerating the discovery of the optimal path.

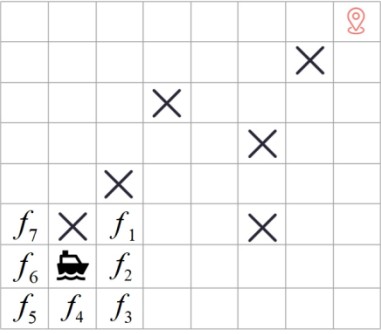

Fig. 11. Simulation diagram of USV accessing nodes

### D. Real-time Obstacle Avoidance in Local Path Planning

In polar unstructured waters, the precision and maintainability of local maps are vital due to variable ice obstacles and limited navigation aids. Grid map construction follows three refinement steps to address these challenges.

- Coordinate transformation: The orientation of the local map is aligned with the geographic coordinate system as given by global coordinates (see "Fig. 13(a)"), whereas sensor information is based on the sensor coordinate system (see "Fig. 13(b)"). Hence, it is necessary to

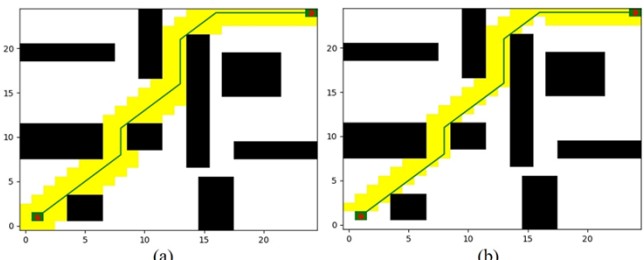

Fig. 12. Simulation diagram of USV accessing nodes

convert the sensor's perception from the sensor coordinate system to the local Cartesian coordinate system.
The transformation is depicted in "equation (14)"

$$\begin{cases} X = x - R \cdot \sin(\alpha - \theta) \\ Y = y + R \cdot \cos(\alpha - \theta) \end{cases} \quad (14)$$

, and "Fig. 13(c)" illustrates the conversion results.
Where $\alpha$ denotes the vehicle's heading angle, $(R, \theta)$ represents the polar coordinates of the target point output by the sensor, and $(x, y)$ is the coordinate position of the target point in the local Cartesian coordinate system.

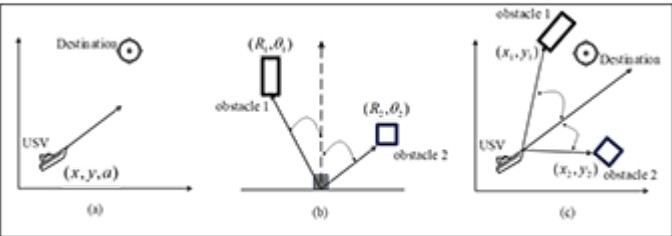

Fig. 13. Coordinate transformation processing: (a) Global Coordinate System, (b) Sensor Coordinate System, (c) Coordinate System Transformation Results

- The correspondence between the element positions in the grid matrix and the coordinates in the local map is shown in "equation (15)".
  $n$ is the column index of the element in the grid matrix; $m$ is the row index in the grid matrix; $p$ is the grid map resolution, the actual distance per grid cell; $x$ and $y$ are the horizontal and vertical coordinates of the target in the actual map, respectively, with $y_{\max}$ denoting the maximum vertical coordinate of the local map.

$$\begin{cases} n = floor(\frac{x}{p}) \\ m = floor(\frac{y_{max} - y}{p}) \end{cases} \quad (15)$$

- The configuration of redundant empty collision spaces aims to reserve a safety margin between the planned path and obstacles, ensuring the setup for unmanned vessels. Implementing redundant collision spaces on a grid map involves the following key steps:
  1.Establishing the maximum distance $D$ between the center of the unmanned vessel's path trajectory and its contour is intended to precisely quantify the greatest distance from the trajectory center to the vessel's perimeter

during navigation, serving as a theoretical foundation for the vessel's path planning and collision avoidance strategies.
2.Ensure precise correspondence between spatial resolution and actual physical distance; the actual distance represented by a single grid side length is calculated according to the following formula to determine the minimum number of grids to be retained.
3.In the process of grid map coarsening, the directly adjacent grids around obstacles (i.e., above, below, to the left, and to the right) are set to impassable.

## IV. RESULTS ANALYSIS AND DISCUSSION

At the current research stage, the efficacy of the GI-ACO-A* algorithm has only been validated in simplified scenarios. Therefore, this section will conduct global path planning experiments using the A* algorithm, ACO-A* algorithm, and GI-ACO-A* on preprocessed radar images (as shown in "Fig. 7") and perform a comparative analysis. This exercise aims to further substantiate that, under complex environmental conditions, the GI-ACO-A* algorithm achieves a significant improvement in execution efficiency compared to the traditional A* algorithm.

### A. Application of the A* Path Planning Algorithm in Rasterized Radar Image Scenarios

"Fig. 14(a)" illustrates the initial layout for global path planning with Node A as the starting position and Node B as the target, while "Fig. 14(b)" depicts the set of nodes traversed by the A* algorithm during path exploration, with the yellow area indicating the visited nodes.

It is evident from Figure 21 that the path generated by the A* algorithm exhibits high smoothness. However, the efficiency is significantly impacted by the extensive node visits during execution. Specifically, the algorithm traversed a total of 25,374 nodes and took 33.119 seconds to complete.

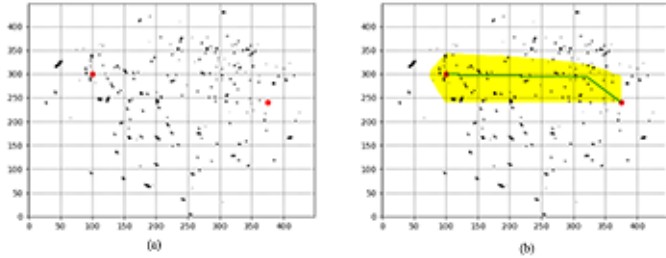

Fig. 14. Figure of A* Path Planning Algorithm Application Results

### B. Application of the ACO-A* Path Planning Algorithm in Rasterized Radar Image Scenarios

"Fig. 15(a)" presents the optimized pheromone concentration update mechanism, which significantly enhances the efficiency of the A* algorithm in the node search process through initial assignment to each grid cell. This improved formula has a significant impact on the node evaluation and selection

strategy during the path optimization process. "Fig. 15(b)" shows the set of nodes traversed by the ACO-A* algorithm during path exploration, with the yellow area indicating the visited nodes.

As observed in "Fig. 15(b)" compared to "Fig. 14(b)", the number of visited nodes (indicated by the yellow area) is significantly reduced to 10,439 nodes. Concurrently, the execution time of the algorithm is correspondingly reduced to 2.902 seconds. This demonstrates that the A* algorithm with Ant Colony Optimization (ACO) strategy significantly improves the execution efficiency, achieving more effective node search and path optimization in path planning tasks.

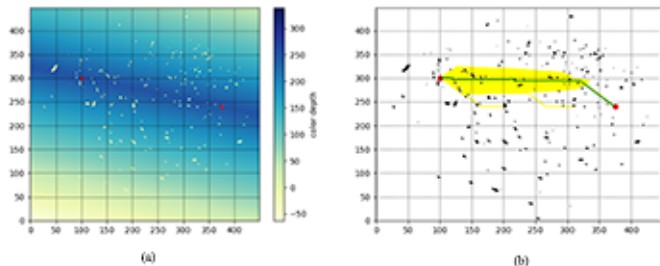

Fig. 15. Figure of ACO-A* Path Planning Algorithm Application Results

### C. Application of the GI-ACO-A* Path Planning Algorithm in Rasterized Radar Image Scenarios

"Fig. 16(a)" enhances the A* algorithm by optimizing the heuristic function values based on the directional position of target B relative to starting point A, prioritizing directional grid selection and thereby reducing the search space, which significantly increases the algorithm's efficiency. "Fig. 16(b)" illustrates the node set traversed by the GI-ACO-A* algorithm during path exploration, with the yellow area denoting the visited nodes.

"Fig. 16(b)" clearly demonstrates a notable decrease in the number of nodes visited, as indicated by the yellow area, to only 2,072 nodes. This reduction correspondingly improves the algorithm's efficiency, with the time taken reduced to just 0.223 seconds. Compared to the ACO-A* algorithm, this refined algorithm achieves a substantial reduction in execution time, underscoring its superior efficiency in path search.

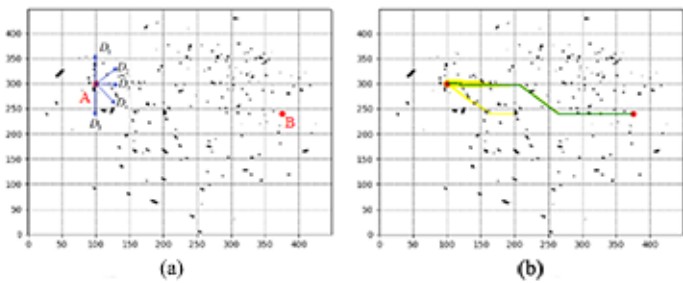

Fig. 16. Figure of GI-ACO-A* Path Planning Algorithm Application Results

### D. Real-time obstacle avoidance for local path planning.

In the local scenario depicted in "Fig. 17(a)", the unmanned surface vessel approaches an area with obstacles, where the risk of potential collisions between its contour and the ice layer is high. Although a theoretically optimal path to the operational site can be found within the global path planning framework, the vessel still faces the risk of colliding with ice layers in local areas during actual navigation. To effectively mitigate such risks, a redundant space setting is introduced to enhance the safety of the path planning strategy.

As shown in "Fig. 17(b)", the high continuity regions of the ice layer (represented by the extensive black obstacle areas) signify high-risk zones for navigation. To improve the efficacy of the path planning algorithm, this study incorporates scattered sea ice units around continuous ice layers into a predefined redundant space for optimized path generation. This ensures that the resulting paths not only maintain high smoothness but also strictly adhere to the kinematic constraints of the unmanned surface vessel, thereby enhancing navigation safety and efficiency.

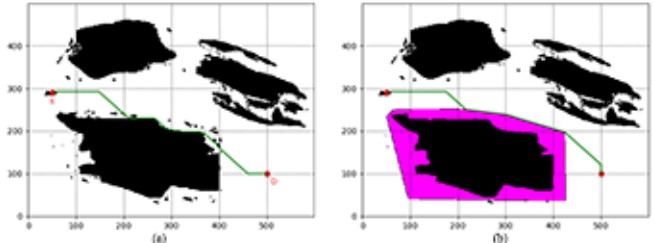

Fig. 17. Local path planning route map (after optimization)

## V. CONCLUSION

This paper introduces a rapid path planning method for polar surface unmanned vessels based on the GI-ACO-A* algorithm, aiming to address the inefficiencies of traditional A* algorithms in path search and to integrate global navigation with local planning in unstructured environments, significantly enhancing the efficiency and safety of path planning.

In terms of environmental modeling and rasterization: The Curvature-Assisted Circle Detection algorithm (CACD) and an improved Emperor Penguin Optimization algorithm are utilized for ROI extraction and sea ice distribution filtering in radar images. Non-sea ice pixel filtering is further achieved through a K-means-based RGB clustering classification algorithm, facilitating image rasterization.

Regarding the comparison and analysis of path planning algorithms: Simulation tests were conducted to compare the performance of the A*, ACO-A*, and GI-ACO-A* algorithms in rasterized radar image scenarios. The results indicate that the GI-ACO-A* algorithm significantly outperforms the other two in terms of node visits and execution time, reducing node visits from 25,374 to 2,072 and execution time from 33.119 seconds to 0.223 seconds, demonstrating its efficiency in complex environments.

For local path planning and real-time obstacle avoidance: In the unstructured polar water surface environment, the introduction of redundant space enhances the safety of the path planning strategy. Real-time obstacle avoidance through local path planning ensures effective collision risk mitigation as the unmanned vessel approaches obstacle areas.

The proposed GI-ACO-A* algorithm and its navigation strategy exhibit significant advantages in polar surface unmanned vessel path planning, providing robust technical support for the efficient operation of unmanned vessels in complex environments.

## VI. Acknowledgement

This work was supported in part by the National Natural Science Foundation of China under Grant 52331012, Grant 52071200, Grant 52201401, Grant 52201403, and Grant 52102397; in part by the National Key Research and Development Program of China under Grant 2021YFC2801002; in part by the Shanghai Committee of Science and Technology, China, under Grant 23010502000.

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
