# OpenReview forum: "Fast Path Planning for Polar Surface Unmanned Vessels Based on GI-ACO-A* Algorithm"
_IEEE.org/ICIST/2024/Conference — IEEE ICIST 2024 Conference Submission_

### Official Review · Reviewer_mkdu · 2024-08-23
**This paper an be accepted after minor modifications.**

**Rating:** 7
**Confidence:** 3

**Review:**

This paper proposes the GI-ACO-A* algorithm (Goal-Induced A* under Ant Colony Optimization influence), along with a navigation strategy integrating global path planning with local obstacle avoidance for unstructured environments, which features a global layer for overview path planning and a local layer for real-time obstacle evasion, adapting to the complex and dynamic surface conditions. Finally, simulation tests have validated its navigation safety and efficiency under complex surface conditions.
1). The contributions should be stressed more in comparison with specific existing works.
2). The quality of Fig. 5 is not clear and needs to be improved.

---

### Official Review · Reviewer_jt6T · 2024-08-24
**The paper is written clearly, exceptionally excellent.**

**Rating:** 8
**Confidence:** 3

**Review:**

This paper excels in terms of quality, clarity, originality, and significance, but I would still like to offer some suggestions.
1. Please describe your future work.
2. Can the method proposed in this paper be applied to actual systems?

---

### Official Review · Reviewer_r4db · 2024-08-25
**The manuscript is well-organized and clearly stated.**

**Rating:** 9
**Confidence:** 4

**Review:**

In the manuscript titled"Fast Path Planning for Polar Surface Unmanned Vessels Based on GI-ACO-A* Algorithm"proposes the GI-ACO-A* algorithm, which enhances path search efficiency for unmanned surface vessels by reducing redundant node visits and incorporating vessel profile considerations. The algorithm integrates global path planning with local obstacle avoidance and processes radar imagery to create a sea ice distribution map, demonstrating improved navigation safety and efficiency in complex environments.This work provides new insight into the development of efficient path-finding algorithms for unmanned surface vessels by integrating global and local navigation strategies and leveraging radar imagery for environmental mapping.The manuscript is well-organized and clearly stated.

---

### Decision · Program_Chairs · 2024-09-08

Accept (Oral)